# Inverted pendulum driven by a horizontal random force: statistics of the never-falling trajectory and supersymmetry

Nikolai A. Stepanov and Mikhail A. Skvortsov⋆

L. D. Landau Institute for Theoretical Physics, Chernogolovka 142432, Russia
⋆ skvor@itp.ac.ru

September 23, 2021

## Abstract

We study stochastic dynamics of an inverted pendulum subject to a random force in the horizontal direction (stochastic Whitney's problem). Considered on the entire time axis, the problem admits a unique solution that always remains in the upper half plane. Assuming a white-noise driving, we develop a field-theoretical approach to statistical description of this never-falling trajectory based on the supersymmetric formalism of Parisi and Sourlas. The emerging mathematical structure is similar to that of the Fokker-Planck equation, which however is written for the "square root" of the probability distribution function. An exact analytical solution is obtained in the limit of strong driving.

# 1   Introduction

A dynamic system driven by a time-dependent perturbation generically demonstrates diffusion in the energy space. A paradigmatic example showing that sort of behavior is the kicked classical rotator, a particle on a ring subject to position-dependent periodic kicks [1]. When the kicks' strength exceeds a certain threshold value, rotator's time evolution described by the standard map [2] becomes chaotic, and since the spectrum is bounded from below, energy-space diffusion translates to the linear growth of the average kinetic energy with time. Such a behavior is not specific to classical physics with one degree of freedom. The same phenomenon also takes place, for example, for quantum systems of many fermions. Provided such a system can be described by the random matrix theory [3] (e.g., in a quantum dot geometry [4]), one finds that under the action of a time-dependent perturbation the fermionic distribution function in the energy space evolves according to the diffusion equation [5,6]. Due to the Fermi statistics, this leads to the growth of the system energy, eventually leading to dissipation.

    Diffusion in the energy space and associated growth of the system energy can be suppressed by a peculiar quantum effect known as dynamic localization, when destructive interference between different paths blocks further energy increase. Dynamic localization takes place both for a quantum kicked rotator [7–9] and a many-electron quantum dot under periodic driving [10].

    A very different mechanism of blocking energy diffusion in a pumped mechanical system takes place if there exists a very specific trajectory, which remains localized in a bounded region of phase space during the entire motion. A famous example is a driven inverted pendulum (see inset to Fig. 1) described by the equation

$$\ddot{\theta} = \omega^2 \sin\theta + f(t)\cos\theta, \tag{1}$$

where $\theta(t)$ is the angle of the pendulum counted from the upward position, and $f(t)$ is a random force acting in the horizontal direction. A *typical* trajectory starting in an upper half-plane will deviate exponentially from the upright position and go to the lower half-plane to minimize the potential energy. Later on it will exhibit chaotic motion with many rotations around the pivot point, gradually increasing its average total energy.

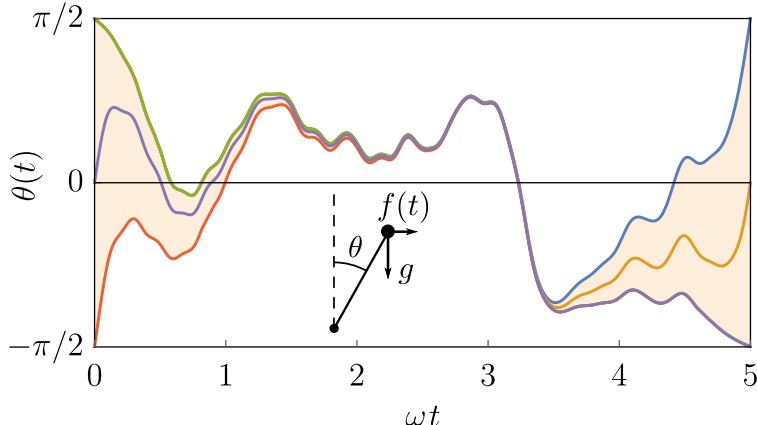

Figure 1: Examples of non-falling solutions to Eq. (1) bounded to the strip $|\theta(t)| < \pi/2$ for a particular realization of the drive $f(t)$ at the interval $(0, T)$ with $\omega T = 5$. For any choice of $\theta_1$ and $\theta_2$ within the strip, there exists a unique trajectory satisfying the boundary conditions $\theta(0) = \theta_1$ and $\theta(T) = \theta_2$. In the limit $T \to \infty$, the bundle of these trajectories (shown by filling) becomes infinitely thin, thus defining a unique *never-falling trajectory*, whose statistical properties are studied in our paper. Inset: sketch of a driven inverted pendulum.

Remarkably, for each driving force $f(t)$ there exist a special *non-falling trajectory* (non-FT), which always remains in the upper half plane, $-\pi/2 < \theta(t) < \pi/2$. In mathematics, existence of a non-FT for Eq. (1) was first addressed by Courant and Robbins (CR) in the book "What is mathematics?" published in 1941 [11], where the problem was attributed to Whitney. Their proof of the existence was based on the intermediate value theorem and essentially relied on the assumptions of a continuous dependence of the final pendulum position on initial conditions. Lack of rigor in the original arguments of CR stimulated a long-lasting discussion in mathematical literature (for a review, see Refs. [12] and [13]), and the very existence of the non-FT was questioned [14]. An important refinement of the CR proof was made by Broman [15], who utilized the fact that the sets of initial conditions leading to touching the left ($\theta = -\pi/2$) or right ($\theta = \pi/2$) boundary are open. Nevertheless, in 2002, Arnold considered this problem still open [16].

Arnold's comment triggered a new wave of interest in Whitney's problem. In 2014, Polekhin gave a proof [17] based on the topological Wazewski principle. Polekhin's work was followed by a number of publications, where his approach was generalized and new topological methods to prove existence of the non-FT were applied [12, 18, 19].

To illustrate the concept of the non-FT, in Fig. 1 we plot nine non-falling solutions of the boundary-value problem for the pendulum equation (1) with $\theta(0) = -\pi/2, 0, \pi/2$ and $\theta(T) = -\pi/2, 0, \pi/2$ calculated for the same given realization of the driving force $f(t)$ on the interval $(0, T)$. Each trajectory is obtained by adjusting the initial velocity $\dot{\theta}(0)$ to keep the trajectory in the strip $|\theta(t)| < \pi/2$. In accordance with CR, a non-falling solution of the boundary-value problem exists for any initial and final value within the strip.

A crucial observation that we make is that *the non-FT solving the boundary-value problem for Eq. (1) is unique*. To the best of our knowledge, the uniqueness of non-FT has not been discussed in the mathematical literature. On the one hand, this fact can be easily verified by direct numerical simulations. On the other hand, it can be proved with the help of the lemma claiming that if two non-FT $\theta_1(t)$ and $\theta_2(t)$ are such that $\theta_1(\tau) < \theta_2(\tau)$ and $\dot{\theta}_1(\tau) < \dot{\theta}_2(\tau)$

then $\theta_1(t) < \theta_2(t)$ for all $t > \tau$. A detailed proof of the uniqueness of the non-FT will be given elsewhere.

The shaded region in Fig. 1 shows a bundle of non-falling solution for the boundary-value problem with all possible start and end points. For a sufficiently long time interval ($\omega T \gg 1$), these trajectories diverge significantly only near the end points, whereas at intermediate values of $t$ they closely follow each other. The larger is $T$, the smaller is the width of the bundle. Finally, if we extend the time interval at which we study non-falling solutions to the whole real axis, the bundle of non-falling trajectories will become infinitely thin, thus defining *a unique never-falling trajectory*. The never-falling trajectory is an attractor of all non-falling trajectories defined on a finite time interval. This attractor is absolutely unstable: any deviation from it will exponentially quickly take the trajectory out of the strip $-\pi/2 < \theta < \pi/2$.

A never-falling trajectory (hereafter denoted by NFT) is a complicated functional of the driving force $f(t)$. Obtaining it for a given $f(t)$ is equivalent to solving an inverse control problem in control theory [20]. However when the pendulum is driven by an irregular force (noise), instead of restoration of the particular form of the NFT, it is more natural to address its statistical properties.

## 2 Statistical properties of the never-falling trajectory

In this Letter, we analyze the statistics of the never falling trajectory in the limiting case of a random driving described by the white-noise correlation function

$$\langle f(t)f(t') \rangle = 2\alpha\delta(t-t'). \tag{2}$$

The model (2) applies when the correlation time of $f(t)$ is much shorter than the pendulum's oscillation period, $2\pi/\omega$. Then NFT statistics depend on the single dimensionless parameter $\alpha/\omega^3$.

### 2.1 Analytical solution of the linearized problem

As a warm-up, consider the simplest case of weak driving, $\alpha \ll \omega^3$, when the angle at the NFT remains small and Eq. (1) can be linearized. Then we obtain a *linear* problem with an *additive* noise, $\ddot{\theta} = \omega^2\theta + f(t)$, which can be immediately solved with the Green's function method. The requirement that the trajectory stay near the origin dictates the choice of the Feynman Green's function $G_{\mathrm{F}}(t) = -\exp(-\omega|t|)/2\omega$, which decays both in the future and in the past. The choice of the Feynman Green's function—neither retarded nor advanced—reflects the peculiarity of the problem, which is not of evolutionary type. In this way one obtains an explicit expression for the NFT as a functional of the driving force: $\theta(t) = \int G_{\mathrm{F}}(t-t')f(t')\,dt'$. We see that indeed the NFT is uniquely defined for a given driving $f(t)$. Finally, averaging over the white noise (2), we get a Gaussian probability distribution function (PDF) of the instantaneous coordinate $\theta$ and momentum $p = d\theta/dt$:

$$P(\theta, p) = \frac{\omega^2}{\pi\alpha}\exp\left(-\frac{\omega^3}{\alpha}\theta^2 - \frac{\omega}{\alpha}p^2\right). \tag{3}$$

An approximate expression (3) is valid at $\alpha/\omega^3 \ll 1$, when the angle $\theta(t)$ is typically small and the NFT does not reach the boundaries $\theta = \pm\pi/2$. At larger driving the nonlinearity of the equation of motion (1) becomes important, and the explicit construction of the NFT for a given $f(t)$ seems impossible. Therefore in order to address the statistics of the NFT at arbitrary $\alpha/\omega^3$ one has to use a different technique that does not rely on exact solution of Eq. (1) but is able to perform disorder averaging at the initial stage of the consideration. One might think that the

suitable approach would be that of the Fokker-Planck (FP) equation for the probability density $P(\theta, p)$ [21]. However it cannot be used to describe the NFT for the following reasons. First, the FP equation describes the ensemble of trajectories, whereas the NFT is a unique trajectory ("of measure zero"). Second, the FP equation belongs to the evolutionary type, while the NFT keeps information on the future behavior of the drive $f(t)$.

## 2.2 Supersymmetric formalism

To attack the problem of the NFT statistics, we suggest to use the sypersymmetric formalism developed by Parisi and Sourlas [22,23], which was inspired by the field-theoretical approach to stochastic classical dynamics developed in 1970s [24,25]. The idea of this formalism is to represent summation over solutions of some classical equation of motion, $L(x) = 0$, for a dynamical variable $x$ by the functional integral over all $x$ weighted with the delta-function $\delta[L(x)]$. Then this delta-function is represented as an integral with an exponent over an auxiliary field $\lambda$, while the emerging determinant due to change of variables is written as a functional integral over a pair of Grassmann fields $\overline{\chi}$, $\chi$. As a result, the theory is formulated in terms of a supersymmetric action $S[x, \lambda, \overline{\chi}, \chi]$, which can be easily averaged over disorder. Though specific to stochastic dynamics, the approach of Parisi and Sourlas follows the general philosophy of theoretical description of disordered systems, where the key point is to invent a functional representation (replica [26,27], supersymmetric [4] or Keldysh [28,29]) suitable for disorder averaging.

In order to implement the outlined procedure for the pendulum equation of motion (1), we write the partition function as a functional integral over all trajectories $\theta(t)$:

$$Z = \int \mathcal{D}\theta(t)\, \delta[-\partial_t^2 \theta + F(\theta)]\, \big|\det[-\partial_t^2 + F'(\theta)]\big|, \tag{4}$$

where $F(\theta) = \omega^2 \sin\theta + f(t)\cos\theta$. Following the standard steps [23,30], we introduce a bosonic field $\lambda(t)$ to put the argument of the delta function to the exponent, and a pair of Grassmann fields $\chi(t)$ and $\overline{\chi}(t)$ to represent the determinant. This leads to

$$Z = \int \mathcal{D}\theta(t)\, \mathcal{D}\lambda(t)\, \mathcal{D}\overline{\chi}(t)\, \mathcal{D}\chi(t)\, e^S, \tag{5}$$

with the action $S[\theta, \lambda, \overline{\chi}, \chi]$ given by

$$S = \int dt \left\{ i\lambda[-\partial_t^2 \theta + F(\theta)] + \overline{\chi}[-\partial_t^2 + F'(\theta)]\chi \right\}. \tag{6}$$

Averaging over the driving force distribution (2) generates an effective action, which can still be written as an integral of a local-in-time Lagrangian due to the white-noise nature of driving, see Appendix A.1.

The key trick to rewriting Eq. (4) in the form of Eq. (5) is to replace the absolute value of the determinant by the determinant itself. This implicitly relies on the assumption that the latter is positive for all solutions of the equation of motion [23,30]. For a generic stochastic equation this is not true, and therefore the Parisi-Sourlas approach cannot be applied as it weights various solutions with arbitrary signs. To work with the absolute value of the determinant one has to resort to much more sophisticated techniques [31,32].

However the problem of the determinant sign does not appear if the solution to the stochastic dynamic equation is unique. This is exactly the case of the NFT for a driven inverted pendulum we are interested in. Hence, it is the uniqueness of the NFT that justifies the use of the Parisi-Sourlas method for the description of its statistics.

## 2.3 Transfer-matrix equation and the probability distribution function

The one-dimensional field theory (5) can be equivalently formulated in the quantum-mechanical language [33], with the transfer-matrix Hamiltonian $\hat{\mathcal{H}}$ acting on the wave function $\hat{\Psi}(\theta, \lambda, \overline{\chi}, \chi)$. Then evaluation of the functional integral is reduced to solving an imaginary-time Schrödinger equation $\partial \hat{\Psi} / \partial t = -\hat{\mathcal{H}}\hat{\Psi}$. The most important circumstance making the statistical description of the NFT possible is its exponentially weak sensitivity to boundary conditions (see Fig. 1). As in the theory of Anderson localization [4], this means that the NFT corresponds to the *zero mode* of the supersymmetric transfer-matrix Hamiltonian: $\hat{\mathcal{H}}\hat{\Psi} = 0$. Singling out the Grassmann content of the wave function,

$$\hat{\Psi}(\theta, \lambda, \overline{\chi}, \chi) = \Psi(\theta, \lambda) + \Phi(\theta, \lambda)\overline{\chi}\chi, \tag{7}$$

we can represent the Hamiltonian as a $2 \times 2$ differential operator acting on the vector $(\Psi, \Phi)$, see Appendix A.2.

As a consequence of the Becchi-Rouet-Stora-Tuytin (BRST) symmetry of the theory [30], there exists a relation between $\Psi$ and $\Phi$, which makes it possible to write an equation for a single function. In the present case, this reduction has the form $\Phi = -i\partial_\theta \Psi / \lambda$, leading to the following equation:

$$\left(\lambda \partial_\lambda \partial_\theta \lambda^{-1} + \omega^2 \lambda \sin\theta + i\alpha\lambda^2 \cos^2\theta\right)\Psi(\theta, \lambda) = 0. \tag{8}$$

The structure of the differential operator in Eq. (8) suggests switching to a new function $\psi(\theta, \lambda) = i\Psi(\theta, \lambda)/\lambda$, which will be the main object of our theory. We also make Fourier transform from the variable $\lambda$ to its conjugate momentum $p$: $\psi(\theta, \lambda) = \int \psi(\theta, p)e^{ip\lambda}dp$. In terms of the function $\psi(\theta, p)$, Eq. (8) takes the form

$$\left(p\partial_\theta + \omega^2 \sin\theta\, \partial_p - \alpha \cos^2\theta\, \partial_p^2\right)\psi(\theta, p) = 0. \tag{9}$$

Remarkably, Eq. (9) mathematically coincides with the FP equation for stochastic dynamics (1) (in its linearized form also known as Kramers equation [21,34]). An essential difference however is that the time-independent FP equation describes the steady PDF $P(\theta, p)$ *of all trajectories*, whereas Eq. (9) is written for an auxiliary function $\psi(\theta, p)$, which encodes statistics of the *unique* NFT. In order to express its instantaneous PDF $P(\theta, p)$ in terms of $\psi(\theta, p)$, one has to evaluate the integral (4) with the prefactors $\delta[\theta - \theta(t)]\delta[p - d\theta(t)/dt]$. After some algebra with Grassmann numbers one finds (see Appendix A.5):

$$P(\theta, p) = \left\{\psi(\theta, p), \psi(\theta, -p)\right\}_{\theta, p}, \tag{10}$$

where $\{f, g\}_{\theta, p} = (\partial_\theta f)(\partial_p g) - (\partial_p f)(\partial_\theta g)$ is the Poisson bracket. The bilinear dependence of the PDF on $\psi$ reflects the fact that the NFT contains knowledge of both the past ($p > 0$) and the future ($p < 0$) [cf. calculation with the Feymann Green's function that lead to Eq. (3)]. A similar bilinear dependence of the single-point wave function correlations on the zero mode of the transfer-matrix Hamiltonian is well known in the theory of quasi-one-dimensional Anderson localization [33, 35, 36]

## 2.4 Boundary conditions

The crucial element of our theory is *boundary conditions* for Eq. (9) that ensure that the trajectory never leaves the region $|\theta(t)| < \pi/2$. It means that the NFT should approach the boundaries $\theta = \pm\pi/2$ with zero velocity: $P(\pi/2, p) = P(-\pi/2, p) = 0$. In accordance with Eq. (10), it suggests that $\psi$ should be constant at the boundary. However since the PDF is

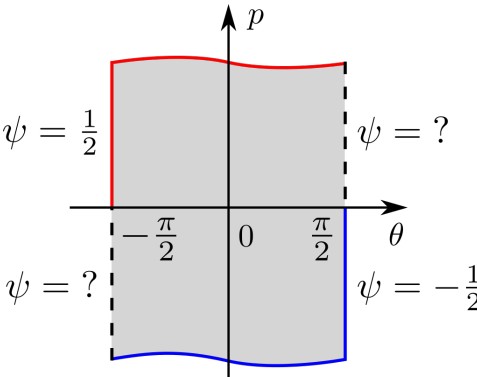

Figure 2: Boundary conditions (11) to Eq. (9) for the function $\psi(\theta, p)$. It should be obtained inside the shaded region and on two dashed segments of the boundary.

bilinear in $\psi$, it is possible to relax this requirement and impose boundary conditions only at the half of the lines $\theta = \pm\pi/2$:

$$\psi(\pi/2, p < 0) = \psi(\theta, -\infty) = -1/2, \tag{11a}$$

$$\psi(-\pi/2, p > 0) = \psi(\theta, \infty) = 1/2. \tag{11b}$$

These boundary conditions are shown in Fig. 2. Since the PDF (10) is expressed in terms of the derivatives of $\psi$, its precise value at the boundary is a matter of convention. However unit normalization of $P(\theta, p)$ imposes a constraint $\psi(\theta, \infty) - \psi(\theta, -\infty) = 1$, see Appendix B. Resolving it in a symmetric way, we arrive at Eqs. (11).

We emphasize that Eqs. (11) do not belong to any known types of boundary conditions to the FP equation discussed in literature (absorbing wall [37], ideally reflecting wall [21], inelastically reflecting wall [38]). All those boundary conditions refer to the standard FP situation when one is interested in forward evolution of an ensemble of trajectories. In contrast, boundary conditions (11) to the FP-like equation (9) describe the behavior of a unique NFT. To some extent, our boundary conditions resemble those for an absorbing wall [37]: both of them do not specify the distribution for outgoing momenta and fix the distribution for incoming momenta. However while an absorbing wall does not transmit particles back, the boundary in Eqs. (11) acts as a source of incoming particles with momentum-independent flux, which has different signs at the opposite parts of the boundary.

The FP equation (9) supplemented by the boundary conditions (11) still constitutes a nontrivial problem due to its non-locality: The function $\psi$ at the part of the boundary, $\psi(\pi/2, p > 0)$ and $\psi(-\pi/2, p < 0)$, should be found simultaneously with the solution of the inner problem. Below we demonstrate that the system of Eqs. (9) and (11) indeed provides a full statistical description of the NFT. In the limiting cases the solution will be obtained analytically, whereas at arbitrary $\alpha/\omega^3$ one should resort to numerical simulations. The results for the function $\psi(\theta, p)$ are shown in Fig. 3 in the limit of weak ($\alpha/\omega^3 \ll 1$) and strong ($\alpha/\omega^3 \gg 1$) driving.

## 3 Results for the probability distribution function

### 3.1 Vanishing driving

In the trivial case of a vanishing driving, $\alpha = 0$, the solution takes a pretty simple form $\psi(\theta, p) = \text{sign}(p - 2\omega \sin \theta/2)/2$. Then two derivatives in the Poisson bracket (10) generate two delta functions in the PDF: $P(\theta, p) = \delta(\theta)\delta(p)$, as expected since the NFT in this case is just the unstable upper position of the pendulum.

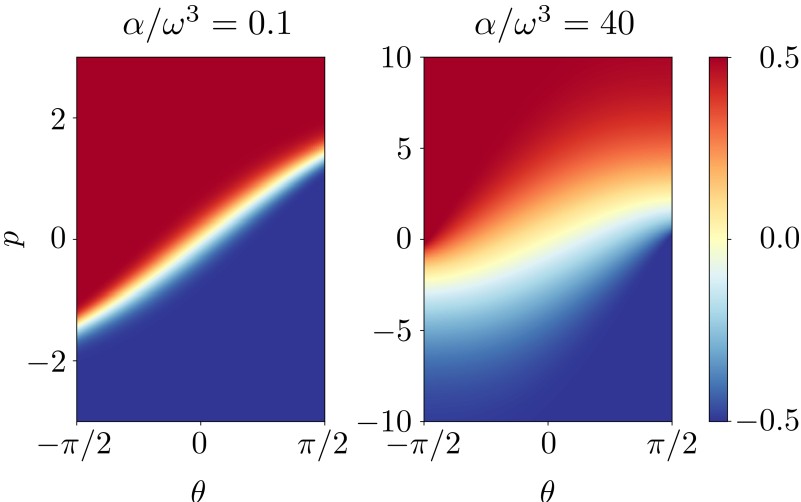

Figure 3: Density plots of $\psi(\theta, p)$ for $\alpha/\omega^3 = 0.1$ (weak driving) and 40 (strong driving) obtained by numerical solution of Eq. (9) with the boundary conditions (11).

## 3.2 Weak driving

In the weak driving limit, $\alpha \ll \omega^3$, the sharp step at the line $p = 2\omega \sin \theta/2$ gets smeared, as can be seen in Fig. 3(a). To find the PDF, which is localized at small angles, the operator in Eq. (9) can be replaced by its linearized version: $p\partial_\theta + \omega^2\theta\partial_p - \alpha\partial_p^2$. The solution then reads $\psi(\theta, p) = \text{erf}[\kappa(p - \omega\theta)]/2$, where $\kappa = (\omega/2\alpha)^{1/2}$. Substituting it into Eq. (10), we recover the weak-noise result (3). Thus we have demonstrated that our approach based on Eq. (9) with the boundary conditions (11) readily reproduces the NFT statistics in the weak-noise limit.

## 3.3 Vanishing vertical force

The other special case when statistics of the NFT can be determined analytically is the limit of a vanishing vertical force, $\omega = 0$ (infinitely strong driving, $\alpha = \infty$). Then one of the three terms in the FP operator in Eq. (9) disappears and the latter can be brought to the canonic form with separated variables:

$$\partial_\tau\psi = q^{-1}\partial_q^2\psi, \tag{12}$$

where $\tau$ and $q$ are new coordinate and momentum defined as $\tau = (2\theta + \sin 2\theta)/\pi$ and $q = (4/\pi\alpha)^{1/3}p$. The boundaries $\theta = \pm\pi/2$ map to $\tau = \pm 1$. The solution of Eq. (12) that satisfies the boundary conditions (11) can be obtained with the help of the multiplicative Airy transform [39] as explained in Appendix C:

$$\psi(\tau, q) = \frac{3\,\text{Ai}'(0)}{\text{Ai}(0)} \int_{-\infty}^{\infty} \frac{d\mu}{\mu}\, \text{Ai}\big[(3/2)^{2/3}\mu^2\big]\text{Ai}(\mu q)e^{\mu^3\tau}. \tag{13}$$

The momentum derivative needed to calculate the PDF is computed in Eq. (55):

$$\partial_q\psi = -\frac{3^{1/6}\,\text{Ai}'(0)}{\text{Ai}(0)}\frac{\text{Ai}(s^2)\exp\left(\frac{2}{3}\tau s^3\right)}{(1-\tau^2)^{1/6}}, \tag{14}$$

where $s = q/[6(1-\tau^2)]^{1/3}$. The other derivative $\partial_\tau\psi$ can be easily obtained from Eq. (12). Owing to the Poisson-bracket structure of Eq. (10), the normalized PDF in the variables $\tau$ and

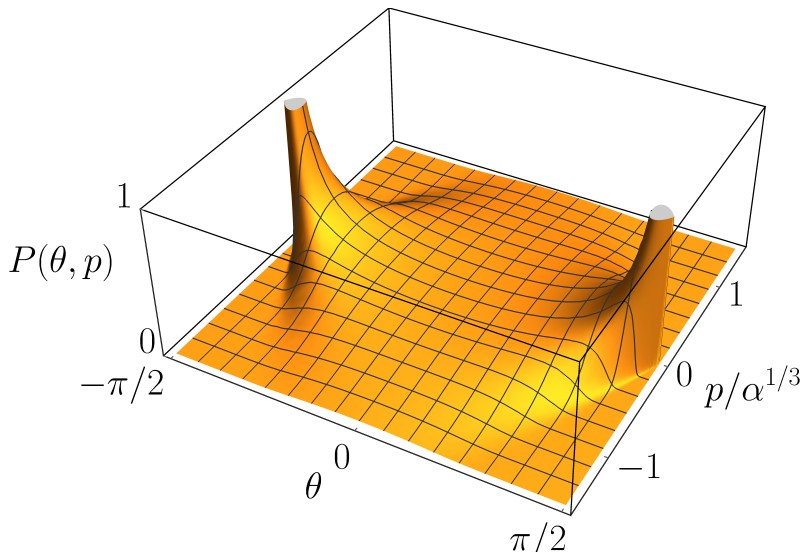

Figure 4: Joint angle and momentum probability distribution function of the NFT, $P(\theta, p)$ in the limit $\omega = 0$, as follows from Eq. (15).

$q$ is given by an analogous expression: $P(\tau, q) = \left\{\psi(\tau, q), \psi(\tau, -q)\right\}_{\tau, q}$, and we arrive at

$$P(\tau, q) = -\frac{2^{4/3}}{3^{1/3}} \left(\frac{\text{Ai}'(0)}{\text{Ai}(0)}\right)^2 \frac{\text{Ai}(s^2)\,\text{Ai}'(s^2)}{1 - \tau^2}. \tag{15}$$

The joint angle and momentum probability distribution function in terms of the original variables given by $P(\theta, p) = (\partial \tau / \partial \theta)(\partial q / \partial p)P(\tau, q)$ is shown in Fig. 4. When $\theta$ approaches the edges at $\pm \pi/2$, the PDF shrinks in the $p$ direction, such that the pendulum touches the horizontal position with zero velocity: $P(\pm \pi/2, p) = 0.338\,\delta(p)$.

Integrating over $q$, we obtain the PDF of the coordinate $\tau$:

$$P(\tau) = \frac{\Gamma(5/6)}{\Gamma(1/3)\Gamma(1/2)} \frac{1}{(1 - \tau^2)^{2/3}}, \tag{16}$$

where $\Gamma$ is the gamma function. The singularities of $P(\tau)$ near the edges ($\tau \to \pm 1$) disappear in the PDF $P(\theta) = (\partial \tau / \partial \theta)P(\tau)$ of the original angle $\theta$:

$$P(\theta) = \frac{4}{\pi^{1/6}} \frac{\Gamma(5/6)}{\Gamma(1/3)} \frac{\cos^2 \theta}{\left[\pi^2 - (2\theta + \sin 2\theta)^2\right]^{2/3}}, \tag{17}$$

which is shown by the solid red line in Fig. 5(d). Surprisingly, $P(\theta)$ is nearly angle-independent, with a minimum 0.303 at the upright position and a maximum 0.338 at the horizontal position of the pendulum ($\theta = \pm \pi/2$).

## 3.4 Arbitrary driving

At arbitrary values of $\alpha/\omega^3$, Eq. (9) with the boundary conditions (11) should be solved numerically. The standard finite element method naturally generalized to include the parts of the boundary with unknown $\psi(\theta, p)$ appears to be stable. Two examples of $\psi(\theta, p)$ obtained numerically at representative values of $\alpha/\omega^3$ are shown in Fig. 3. The resulting PDF of the NFT angle, $P(\theta)$, obtained by integrating $P(\theta, p)$ given by Eq. (10) over $p$ are shown by blue solid lines in Fig. 5.

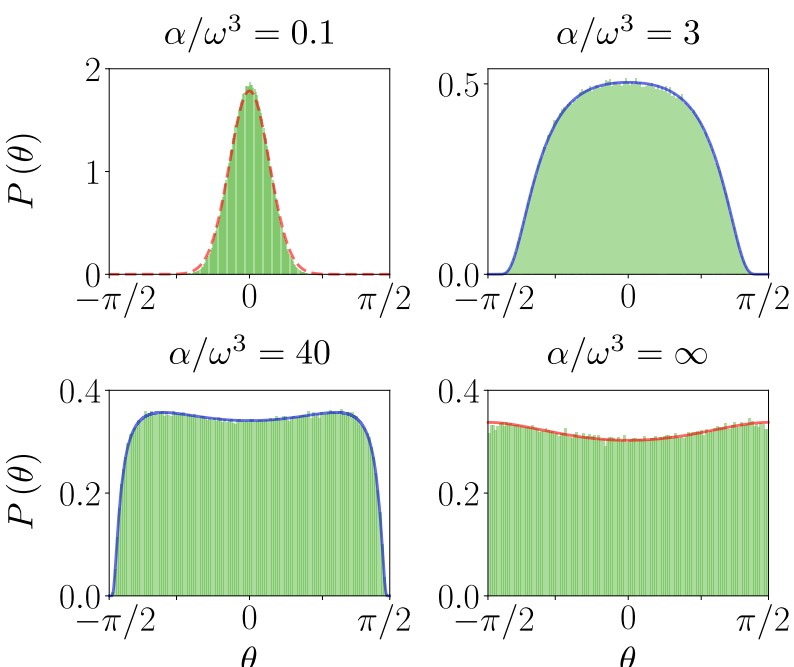

Figure 5: Probability distribution function of the NFT angle, $P(\theta)$, for several values of $\alpha/\omega^3$: (a) 0.1, (b) 3, (c) 40, (d) $\infty$. Green histograms are obtained by direct Monte-Carlo simulation of Eq. (1). The dashed red line in panel (a) is the approximate solution (3). Numerical solutions of Eqs. (9) and (11) are shown by blue lines. The solid red line in panel (d) is the analytical solution (17) for $\omega = 0$.

Figure 5 also demonstrates the results of direct numerical simulation of the NFT from the pendulum equation (1) with randomly generated realizations of $f(t)$. The corresponding PDF histograms are displayed in green. Perfect agreement between $P(\theta)$ obtained from Eqs. (9)–(11) and by direct numerical simulation of Eq. (1) lends strong support for the validity of our theoretical description of the NFT statistics, where disorder averaging is performed at the initial stage of the derivation.

With the increase of the driving strength $\alpha$, the narrow Gaussian distribution (3) shown by the red dashed line in Fig. 5(a) becomes wider, and at $\alpha/\omega^3 \sim 1$ extends almost to the entire interval $(-\pi/2, \pi/2)$. Further increase of $\alpha$ leads to the shrinking of poorly accessible regions near $|\theta| = \pi/2$ and formation of a minimum at $\theta = 0$ (upright position) at $\alpha/\omega^3 \gtrsim 7$. In the limit $\alpha/\omega^3 \to \infty$, the PDF is nearly flat, with 10 % depletion at $\theta = 0$.

## 4 Connections to other mathematical physics problems

### 4.1 Universality of the far-momentum tail of the PDF

Now let us discuss the momentum dependence of the distribution function $P(\theta, p)$ at large $p$. Its Gaussian shape at small $\alpha$ [Eq. (3)] crosses over to the Airy-type behavior (15) in the limit of large $\alpha$, with the large-$p$ asymptotics

$$P_{\text{asymp}}(p) \propto \exp\left[-8|p|^3/9\pi\alpha(1-\tau^2)\right]. \tag{18}$$

Such a form of the far tail is typical for non-linear stochastic problems (e.g., large positive velocity gradient in random-forced Burgers equation [40–42], statistics of extrema in a random

potential [43]) and, as we demonstrate below, is also realized for the driven inverted pendulum at arbitrary $\alpha$. Indeed, searching for the solution of Eq. (9) in the stretch-exponent form, $\psi \propto \exp\left(-g(\theta)|p|^{\beta}\right)$, we see that the term proportional to $\omega^2$ can be neglected at large $|p|$. Hence $\beta = 3$, and using Eq. (10) we arrive at Eq. (18). The asymptotic expression (18) is applicable for $|p| > \omega \cos^{1/2}\theta$, provided that the number in the exponent is large. The stretched-exponential tail of the PDF (18) can be physically understood in terms of an optimal fluctuation of the driving force. To accelerate the pendulum to a large momentum $p$ during the time $\Delta t$, one should exert a force $f \sim p/\Delta t$. The probability $P$ for such a fluctuation can be estimated as $-\ln P \sim f^2 \Delta t/\alpha \sim p^2/\alpha\Delta t$. Since the duration of the pulse is limited by the requirement $|\theta| < \pi/2$, we have $\Delta t \lesssim 1/p$ and hence $-\ln P \sim p^3/\alpha$, in accordance with Eq. (18).

## 4.2   NFT as the global action minimizer

Finally, we suggest looking at the NFT from a different perspective. Consider a boundary value problem for Eq. (1) at a final time interval with $\theta(0) = \theta_1$ and $\theta(T) = \theta_2$, where both the initial and final points are located within the strip $|\theta_{1,2}| < \pi/2$. Numerical simulations demonstrate that the solution to this boundary value problem is not unique if we relax the condition that the trajectory stay within the strip, $|\theta(t)| < \pi/2$. In addition to the NFT that exists for all $T$ (see Fig. 1), other solutions that leave the strip and then come back appear for longer intervals, $T \gtrsim 1/\omega$. Now we ascribe to each solution $\theta(t)$ the value of the corresponding action defined as

$$A[\theta(t)] = \int_{t_1}^{t_2} \left[\dot{\theta}^2/2 - \omega^2 \cos\theta + f(t)\sin\theta\right]dt. \tag{19}$$

Numerical analysis shows that the NFT provides a global minimum for $A[\theta(t)]$ among all solutions of the boundary value problem and therefore is a *minimizer* [41]. This fact provides a connection between the NFT and Burgers equation (whose characteristics are described by the driven pendulum equation) and, more broadly, to the field of one-dimensional turbulence [40, 44, 45].

# 5   Conclusion

To summarize, we introduce a concept of a unique never-falling trajectory for a horizontally driven inverted pendulum and formulate the problem of its statistical description. In the case of white-noise random driving, we provide a full solution for this problem. Using field-theoretical methods of statistical and condensed-matter physics, we express the instantaneous joint PDF of the pendulum's angle and its velocity in terms of an auxiliary function satisfying the Fokker-Planck equation with a new type of boundary conditions. In the limit of very strong driving (vanishing gravitation), the PDF is obtained analytically. For arbitrary driving strength, the derived equations can be easily solved numerically, which is much simpler and efficient than direct numerical simulation of a never-falling trajectory from the equation of motion with statistics accumulation. We demonstrate that both approaches give the same result.

In a wider context, the problem of a never-falling trajectory has many notable connections with other mathematical physics problems: theory of minimizers (NFT is a global minimizer of the classical action), Burgers turbulence, rear events in stochastic differential equations, etc. We expect our approach to be also in demand in control theory. Our results can be naturally generalized to other Langevin-type equations, which admit non-falling trajectories. Finally we'd like to emphasize that studying the properties of a never-falling trajectory, which has measure zero among all solutions of a driven mathematical pendulum, is not just an academic

exercise. For example, the Usadel equation [46] describing inhomogeneous states in dirty superconductors belongs to a class of pendulum equation, but with an essentially complex $\omega$, when balancing an unstable trajectory is a generic situation. The theoretical approach developed in the present publication can then be used to obtain the density of states in inhomogeneous superconducting wires [47].

# Acknowledgements

We thank A. V. Andreev, S. A. Belan, G. Falkovich, M. V. Feigel'man, D. A. Ivanov, I. V. Kolokolov, V. V. Lebedev and I. Yu. Polekhin for stimulating discussions, and I. V. Poboiko for help with numerics.

**Funding information**   This work was supported by the Russian Science Foundation under Grant No. 20-12-00361.

# A   Parisi-Sourlas formalism

## A.1   Supersymmetric functional representation and averaging over driving

Consider a differential equation

$$\partial_t^2 \theta(t) = F(\theta(t)), \tag{20}$$

where $F(\theta)$ might be a nonlinear function with an arbitrary time dependence. For the driven pendulum,

$$F(\theta) = \omega^2 \sin\theta + f(t)\cos\theta. \tag{21}$$

Let $\mathcal{A}[\theta]$ be a functional of a trajectory $\theta(t)$. Then the sum of $\mathcal{A}[\theta]$ over all solutions of the differential equation (20) can be written as a path integral over all functions $\theta(t)$ [22, 23, 30]:

$$\sum_{\text{solutions}} \mathcal{A}[\theta] = \int \mathcal{D}\theta(t)\,\mathcal{A}[\theta(t)]\,\delta[-\partial_t^2\theta + F(\theta)]\,\big|\det[-\partial_t^2 + F'(\theta)]\big|, \tag{22}$$

where the delta function ensures that $\theta(t)$ indeed satisfies the equation, whereas the absolute value of the determinant arises since the argument of the delta function contains the equation rather its solution.

In a generic situation, when Eq. (20) possesses many solutions, the sign of the determinant in Eq. (22) alternates between solutions. However, uniqueness of a non-falling trajectory for Whitney's problem guarantees that the determinant is positive, so that the functional $\mathcal{A}$ on it can be written with the absolute value of the determinant replaced by the determinant itself:

$$\mathcal{A}[\theta_{\text{NFT}}] = \int \mathcal{D}\theta(t)\,\mathcal{A}[\theta(t)]\,\delta[-\partial_t^2\theta + F(\theta)]\det[-\partial_t^2 + F'(\theta)]. \tag{23}$$

Following then the approach of Parisi and Sourlas [22, 23, 30], we rewrite the delta function using an extra field $\lambda(t)$ and the determinant using a pair of Grassmann fields $\overline{\chi}(t)$ and $\chi(t)$:

$$\mathcal{A}[\theta_{\text{NFT}}] = \int \mathcal{D}\theta(t)\mathcal{D}\lambda(t)\mathcal{D}\overline{\chi}(t)\mathcal{D}\chi(t)\,\mathcal{A}[\theta(t)]\,e^S, \tag{24}$$

with the action

$$S = \int L\,dt, \qquad L = i\lambda[-\partial_t^2\theta + F(\theta)] + \overline{\chi}[-\partial_t^2 + F'(\theta)]\chi. \tag{25}$$

Now we average over realizations of a random force $f(t)$, assuming Gaussian white-noise statistics with the correlation functions $\langle f(t)f(t')\rangle = 2\alpha\delta(t-t')$, and integrating by parts in the kinetic term, we obtain

$$\langle \mathcal{A}[\theta_{\mathrm{NFT}}]\rangle = \int \mathcal{D}\theta(t)\,\mathcal{D}\lambda(t)\,\mathcal{D}\overline{\chi}(t)\,\mathcal{D}\chi(t)\,\mathcal{A}[\theta(t)]\exp\left[\int dt\,(i\,\partial_t\lambda\,\partial_t\theta + \partial_t\overline{\chi}\,\partial_t\chi + \mathcal{V})\right], \tag{26}$$

where the superpotential $\mathcal{V}$ is given by

$$\mathcal{V} = \omega^2\left(i\lambda\sin\theta + \overline{\chi}\chi\cos\theta\right) + \alpha\left(i\lambda\cos\theta - \overline{\chi}\chi\sin\theta\right)^2. \tag{27}$$

### A.2 Hamiltonian representation

The one-dimensional field theory (26) can be interpreted as Feynman's path-integral representation of a certain quantum mechanics. An alternative but completely equivalent representation is known to be provided by the time-dependent Schrödinger equation for the wave function $\hat{\Psi}$ with an appropriate transfer-matrix Hamiltonian. This idea of reducing functional integral evaluation to solving a corresponding Shcrödinger equation has appeared to be very fruitful in the theory of quasi-one-dimensional Anderson localization [4, 33]. We exploit the same analogy. In the case of the path integral (26), its evaluation can be reduced to solving the Schrödinger equation

$$\frac{\partial\hat{\Psi}}{\partial t} = -\hat{\mathcal{H}}\hat{\Psi}, \qquad \hat{\mathcal{H}} = -\left(i\partial_\lambda\partial_\theta + \partial_\chi\partial_{\overline{\chi}} + \mathcal{V}\right) \tag{28}$$

for the wave function $\hat{\Psi}(\theta,\lambda,\overline{\chi},\chi)$, which can be expanded in the basis of even elements of the Grassmann algebra according to Eq. (7). In terms of functions $\Psi(\theta,\lambda)$ and $\Phi(\theta,\lambda)$, the Schrödinger equation (28) can be written as

$$\frac{\partial}{\partial t}\begin{pmatrix}\Psi\\\Phi\end{pmatrix} = -\mathcal{H}\begin{pmatrix}\Psi\\\Phi\end{pmatrix}, \qquad \mathcal{H} = -\begin{pmatrix}i\partial_\lambda\partial_\theta + \mathcal{V}_1 & 1\\ \mathcal{V}_2 & i\partial_\lambda\partial_\theta + \mathcal{V}_1\end{pmatrix}, \tag{29}$$

where $\mathcal{V}_{1,2}$ are the coefficients of the expansion of the superpotential (27) over even Grassmann basis: $\mathcal{V} \equiv \mathcal{V}_1 + \mathcal{V}_2\overline{\chi}\chi$.

As in the theory of Anderson localization [4, 33], exponentially weak sensitivity of the NFT to boundary conditions indicates that it corresponds to the *zero mode* of the transfer-matrix Hamiltonian:

$$\begin{pmatrix}i\partial_\lambda\partial_\theta + \mathcal{V}_1 & 1\\ \mathcal{V}_2 & i\partial_\lambda\partial_\theta + \mathcal{V}_1\end{pmatrix}\begin{pmatrix}\Psi\\\Phi\end{pmatrix} = 0. \tag{30}$$

### A.3 Reduction to the scalar equation and Fokker-Planck operator

By construction, the coefficients of the superpotential (27) obey the relation

$$i\lambda\mathcal{V}_2 = \partial_\theta\mathcal{V}_1. \tag{31}$$

As a consequence, the functions $\Psi$ and $\Phi$ become connected by the relation

$$i\lambda\Phi = \partial_\theta\Psi \tag{32}$$

and the system (30) reduces to a single scalar equation for the function $\Psi(\theta,\lambda)$:

$$\left(i\lambda\partial_\lambda\partial_\theta\lambda^{-1} + i\omega^2\lambda\sin\theta - \alpha\lambda^2\cos^2\theta\right)\Psi = 0. \tag{33}$$

It is convenient to introduce a new function

$$\psi(\theta, \lambda) = i\Psi(\theta, \lambda)/\lambda \tag{34}$$

and switch to the momentum representation according to

$$\psi(\theta, \lambda) = \int \psi(\theta, p) e^{ip\lambda} dp. \tag{35}$$

The function $\psi(\theta, p)$ will play a key role in our analysis. In terms of it, the zero-mode equation (33) takes the simplest possible form, coinciding with that of the Fokker-Planck equation:

$$\left( p\partial_\theta + \omega^2 \sin\theta\, \partial_p - \alpha \cos^2\theta\, \partial_p^2 \right) \psi(\theta, p) = 0. \tag{36}$$

At the same time, the functions $\Psi(\theta, p)$ and $\Phi(\theta, p)$ have an elegant representation in terms of the function $\psi(\theta, p)$:

$$\Psi = \partial_p \psi, \quad \Phi = -\partial_\theta \psi. \tag{37}$$

## A.4 BRST symmetry

The relation (32) between $\Psi$ and $\Phi$ that allows to obtain a single equation for the function $\Psi$ is a consequence of the Becchi-Rouet-Stora-Tuytin (BRST) symmetry of the Parisi-Sourlas theory for stochastic differential equations [30]. The Lagrangian of the theory defined in Eq. (25) appears to be invariant with respect to infinitesimal rotations by a Grassmann field $\varepsilon$: $\delta\theta = \varepsilon\chi$ and $\delta\overline{\chi} = -i\varepsilon\lambda$. This means that the Lagrangian satisfies $\hat{\mathcal{D}}L = 0$ and can be written as

$$L = \hat{\mathcal{D}}(\overline{\chi}[-\partial_t^2\theta + F(\theta)]), \tag{38}$$

where $\hat{\mathcal{D}}$ a nilpotent ($\hat{\mathcal{D}}^2 = 0$) BRST operator

$$\hat{\mathcal{D}} = i\lambda\partial_{\overline{\chi}} - \chi\partial_\theta. \tag{39}$$

The BRST symmetry of the Lagrangian translates to the BRST symmetry of the wave functions in the Hamiltonian representation: $\hat{\mathcal{D}}\hat{\Psi} = 0$. Hence there should exist such a function $\psi$ that

$$\hat{\Psi} = \hat{\mathcal{D}}(\overline{\chi}\psi). \tag{40}$$

Comparing with Eqs. (7) and (32), we see that thus defined function $\psi$ coincides (up to an overall sign) with the same function introduced in Sec. A.3.

## A.5 Instantaneous joint PDF of the angle and its velocity

The instantaneous PDF of the angle and momentum of the NFT, $P(\theta, p)$, is defined by Eq. (26) with a local-in-time functional $\mathcal{A}[\theta] = \delta(\theta(t) - \theta)\delta(\dot\theta(t) - p)$. It can be calculated by taking two infinitesimally close moments of time, $t$ and $t + \epsilon$, replacing the functional integrals over the regions $t' < t$ and $t' > t + \epsilon$ by the corresponding zero modes, and discretizing the action at the interval $(t, t + \epsilon)$:

$$P(\theta, p) = \lim_{\epsilon \to 0} \int \frac{d\lambda_1}{2\pi} \frac{d\lambda_2}{2\pi} d\theta_1 d\theta_2 d\chi_1 d\overline{\chi}_1 d\chi_2 \overline{\chi}_2 \, \delta(\theta_1 - \theta) \, \delta\left( \frac{\theta_2 - \theta_1}{\epsilon} - p \right)$$

$$\times \Psi(\lambda_1, \theta_1, \overline{\chi}_1, \chi_1) \exp\left\{ i \frac{(\lambda_2 - \lambda_1)(\theta_2 - \theta_1)}{\epsilon} + \frac{(\overline{\chi}_2 - \overline{\chi}_1)(\chi_2 - \chi_1)}{\epsilon} \right\} \Psi(\lambda_2, \theta_2, \overline{\chi}_2, \chi_2). \tag{41}$$

Substituting $\Psi$ from Eq. (7), performing all integrations and switching to the momentum representation (35), we get

$$P(\theta, p) = \Psi(\theta, p)\Phi(\theta, -p) + \Psi(\theta, -p)\Phi(\theta, p). \qquad (42)$$

In terms of the function $\psi$, the PDF of the NFT takes an amazingly compact form

$$P(\theta, p) = \left\{\psi(\theta, p), \psi(\theta, -p)\right\}_{\theta, p}, \qquad (43)$$

where the Poisson bracket is defined as

$$\{f, g\}_{\theta, p} = (\partial_\theta f)(\partial_p g) - (\partial_p f)(\partial_\theta g). \qquad (44)$$

## B  Boundary conditions

To complete the formulation of the theory for the NFT statistics, we have to impose the boundary conditions on the function $\psi(\theta, p)$ at $\theta = \pm\pi/2$ that will ensure that the pendulum never leaves the upper-half plane. In terms of the PDF, this means that $P(\pm\pi/2, p)$ should vanish for all $p \neq 0$. The structure of Eq. (43) suggests that it is sufficient to nullify $\psi(\pm\pi/2, p)$ not for all $p$, but only on a half-line. We find it convenient to resolve this constraint by imposing the following boundary equations:

$$\psi(\pi/2, p < 0) = -1/2, \qquad (45a)$$
$$\psi(-\pi/2, p > 0) = 1/2. \qquad (45b)$$

As a consequence of the Fokker-Planck equation (36), it follows that

$$\psi(\theta, \pm\infty) = \pm 1/2. \qquad (46)$$

Let us demonstrate that the Fokker-Planck equation (36) with the boundary conditions (45) generates the PDF $P(\theta, p)$, which is automatically normalized to unity. Using Eq. (43) and integrating one of the two terms in the Poisson bracket by parts over $\theta$ and over $p$, we obtain that the bulk contribution vanishes and only the boundary contributes:

$$\int P(\theta, p)\, d\theta\, dp = \int_{-\pi/2}^{\pi/2} d\theta\, \psi(\theta, -p)\, \partial_\theta \psi(\theta, p)\Big|_{p=-\infty}^{p=\infty} - \int_{-\infty}^{\infty} dp\, \psi(\theta, -p)\partial_p \psi(\theta, p)\Big|_{\theta=-\pi/2}^{\theta=\pi/2}. \qquad (47)$$

The first term here vanishes due to Eq. (46), whereas the second term reduces to the integral over two momentum haft-lines due to the boundary conditions (45):

$$\int P(\theta, p)\, d\theta\, dp = -\int_0^\infty dp\, \psi(\pi/2, -p)\, \partial_p \psi(\pi/2, p) + \int_{-\infty}^0 dp\, \psi(-\pi/2, -p)\, \partial_p \psi(-\pi/2, p). \qquad (48)$$

Here $\psi(\pm\pi/2, -p)$ are constants given by $\psi(\theta, \mp\infty) = \pm 1/2$, so it remains to integrate full derivatives. Using the continuity of $\psi(\theta, p)$, we arrive at

$$\int P(\theta, p)\, d\theta\, dp = [\psi(\theta, \infty) - \psi(\theta, -\infty)]^2 = 1, \qquad (49)$$

that proves proper normalization of the PDF.

Finally, we discuss the degrees of freedom in defining the boundary conditions (45). The first one is related to the fact that the function $\psi(\theta, p)$ enters physical observables only via its

derivatives [see Eqs. (37) and (43)]. Therefore it is defined up to an additive constant. Normalization of the PDF ensures that $\psi(\theta, \infty) - \psi(\theta, -\infty) = 1$. In our boundary conditions (45), we resolve this constraint in a symmetric way. The second uncertainty is related to the choice of half-lines of momentum, where $\psi(\pm\pi/2, p)$ should be constant. Besides the convention (45), there is an alternative way to fix the values of $\psi(\pi/2, p > 0)$ and $\psi(-\pi/2, p < 0)$ on the different half-lines of $p$ that would correspond to changing the sign of the momentum. However since the PDF is a bilinear function of $\psi(\theta, p)$ and $\psi(\theta, -p)$, such a choice would lead to the same expression for the PDF.

## C   Exact solution in the absence of gravitation ($\omega = 0$)

### C.1   Expansion in the Airy functions

Remarkably, in the absence of a vertical force, the PDF of the NFT can be obtained exactly. In this limit, the Fokker-Planck equation (36) contains only two terms that allows to separate the variables. To this end we define a new coordinate

$$\tau = \frac{4}{\pi} \int_0^\theta \cos^2\theta' d\theta' = \frac{2\theta + \sin 2\theta}{\pi}, \tag{50}$$

where the overall numerical coefficient is chosen such that the boundaries $\theta = \pm\pi/2$ are mapped to $\tau = \pm 1$, and a new momentum

$$q = (4/\pi\alpha)^{1/3} p. \tag{51}$$

In terms of $\psi(\tau, q)$, Eq. (36) takes a simple form:

$$\partial_\tau \psi = q^{-1} \partial_q^2 \psi. \tag{52}$$

The eigenfunctions of the operator at the right-hand side of Eq. (52) are the Airy functions $\phi_\mu(q) = \text{Ai}(\mu q)$ [48] labeled by a continuous index $\mu \in \mathbb{R}$, with the corresponding eigenvalues $\varepsilon_\mu = \mu^3$. Therefore a general solution of Eq. (52) can be expressed as

$$\psi(\tau, q) = \int_{-\infty}^{\infty} d\mu\, c(\mu) \text{Ai}(\mu q) \exp(\mu^3 \tau), \tag{53}$$

where $c(\mu)$ are the coefficients of the multiplicative Airy transform [39] of the function $\psi$ at the line $\tau = 0$ (upper pendulum position).

### C.2   Solution for $c(\mu)$

The function $\psi(\tau, q)$ written in terms of the integral representation (53) solves the Fokker-Planck equation (52) in the inner region, $|\tau| < 1$ (i.e., $|\theta| < \pi/2$). An unknown function $c(\mu)$ should be determined from the boundary conditions (45) at $|\theta| = \pi/2$. Here we demonstrate that the proper $c(\mu)$ is given by

$$c(\mu) = \frac{3\,\text{Ai}'(0)}{\text{Ai}(0)} \frac{\text{Ai}\big[(3/2)^{2/3}\mu^2\big]}{\mu}. \tag{54}$$

The idea behind the ansatz (54) is that the contribution of large $\mu$ tends to explode at large $|\tau|$ due to the exponential factor in Eq. (53) and unboundedness of the spectrum $\varepsilon_\mu = \mu^3$. Therefore to make the integral (53) convergent at $|\tau| \leq 1$, the coefficients $c(\mu)$ should decay

not slower than $\exp(-|\mu|^3)$. At large $|\mu|$, the function (54) indeed decays sufficiently fast: $c(\mu) \sim \text{sign}\,\mu \exp(-|\mu|^3)/|\mu|^{3/2}$ [see Eq. (66)]. Right at the boundary, $|\tau| = 1$, the leading growing and decreasing exponents fully compensate each other.

To prove that $\psi(\tau, q)$ in the form (53) with $c(\mu)$ given by Eq. (54) satisfies the boundary conditions (45), we show below (i) that $\psi(\pm 1, q)$ has a constant value for $q < 0$ ($q > 0$) and (ii) that its asymptotics at $q \to \pm\infty$ is given by $\frac{1}{2}\,\text{sign}\,q$.

The first statement can be proved by calculating the momentum derivative of $\psi$ with the help of Eq. (69):

$$\partial_q \psi(\tau, q) = -\frac{3^{1/6}\,\text{Ai}'(0)}{\text{Ai}(0)} \frac{\text{Ai}(s^2)\exp\left(\frac{2}{3}\tau s^3\right)}{(1 - \tau^2)^{1/6}}, \tag{55}$$

where we introduced a short-hand notation

$$s = \frac{q}{[6(1 - \tau^2)]^{1/3}} = \frac{2^{1/3}p}{[3\pi\alpha(1 - \tau^2)]^{1/3}}. \tag{56}$$

Talking the limit $|\tau| \to 1$ using the asymptotic expansion (66) and the identities (65), we find

$$\partial_q \psi(\pm 1, q) = \frac{3^{2/3}}{2^{1/6}\Gamma(1/6)} \frac{\exp\left(-|q|^3/18\right)}{|q|^{1/2}} \theta(\pm q). \tag{57}$$

Thus we establish that $\partial_q \psi(\pm 1, q)$ vanishes for $q < 0$ ($q > 0$), in accordance with the boundary conditions (45).

The second statement about normalization is proved by considering the asymptotic behavior of $\psi(\tau, q)$ at $q \to \pm\infty$. In this limit, the main contribution to the integral (53) comes from small $\mu \sim 1/q$ that allows to take all other functions at $\mu = 0$. Using Eq. (67), we obtain:

$$\lim_{q \to \infty} \psi(\tau, q) = 3\,\text{Ai}'(0) \int d\mu \frac{\text{Ai}(\mu q)}{\mu} = 3\,\text{Ai}'(0)\,\text{sign}\,q \int d\mu \frac{\text{Ai}(\mu)}{\mu} = \frac{\text{sign}\,q}{2}, \tag{58}$$

as prescribed by the boundary conditions (45). This completes the proof that Eqs. (53) and Eq. (54) provide an exact analytic solution for the function $\psi(\tau, q)$ in the limit of vanishing $\omega$.

## C.3   Probability distribution function

The joint angle and momentum PDF $P(\theta, p)$ is given by Eq. (43). Owing to the Poisson-bracket structure of this equation, the normalized PDF in the variables $\tau$ and $q$ [see Eq. (50) and (51)] is given by an analogous expression:

$$P(\tau, q) = \left\{\psi(\tau, q), \psi(\tau, -q)\right\}_{\tau, q}. \tag{59}$$

Taking the $q$-derivative from Eq. (55) and calculating then the $p$-derivative from Eq. (52), we arrive at the following expression for $P(\tau, q)$:

$$P(\tau, q) = -\frac{2^{4/3}}{3^{1/3}} \left(\frac{\text{Ai}'(0)}{\text{Ai}(0)}\right)^2 \frac{\text{Ai}(s^2)\,\text{Ai}'(s^2)}{1 - \tau^2}, \tag{60}$$

where $s$ is defined in Eq. (56). The PDF in terms of the original variables $\theta$ and $p$ is given by

$$P(\theta, p) = \frac{\partial \tau}{\partial \theta} \frac{\partial q}{\partial p} P(\tau, q) = -\frac{16\cos^2\theta}{3^{1/3}\pi^{4/3}\alpha^{1/3}} \left(\frac{\text{Ai}'(0)}{\text{Ai}(0)}\right)^2 \frac{\text{Ai}\left(s^2\right)\text{Ai}'\left(s^2\right)}{1 - \tau^2}. \tag{61}$$

The PDF for the coordinate is obtained by integration over the corresponding momentum, which is done with the help of Eq. (68). As a result, the PDF of the variable $\tau$ takes a simple form:

$$P(\tau) = \frac{\Gamma(5/6)}{\Gamma(1/2)\Gamma(1/3)} \frac{1}{(1-\tau^2)^{2/3}}. \tag{62}$$

In the original $\theta$ representation we obtain

$$P(\theta) = \frac{4}{\pi^{1/6}} \frac{\Gamma(5/6)}{\Gamma(1/3)} \frac{\cos^2\theta}{[\pi^2 - (2\theta + \sin 2\theta)^2]^{2/3}}, \tag{63}$$

## D   Mathematical supplementary: Integrals with the Airy functions

### D.1   Basic identities and integrals for the Airy functions

Here we collect useful identities for the Airy function that are needed to describe the solution of the pendulum problem in the limit $\omega = 0$. Many facts about the Airy function can be found in a comprehensive monograph [48].

Integral representation for the Airy function:

$$\text{Ai}(x) = \int_{-\infty}^{\infty} \frac{dt}{2\pi} \exp\left(i\frac{t^3}{3} + ixt\right). \tag{64}$$

Airy function and its derivative at the origin are given by:

$$\text{Ai}(0) = \frac{1}{3^{2/3}\Gamma(2/3)}, \quad \text{Ai}'(0) = -\frac{1}{3^{1/3}\Gamma(1/3)}. \tag{65}$$

Asymptotic expansion at $x \gg 1$:

$$\text{Ai}(x) \approx \frac{1}{2\sqrt{\pi}} \frac{\exp\left(-\frac{2}{3}x^{3/2}\right)}{x^{1/4}}. \tag{66}$$

Useful integrals:

$$\int dx \frac{\text{Ai}(x)}{x} = \int_0^{\infty} dx \frac{\text{Ai}(x) - \text{Ai}(-x)}{x} = \frac{1}{6\,\text{Ai}'(0)}, \tag{67}$$

$$\int_{-\infty}^{\infty} dx\, \text{Ai}(x^2)\text{Ai}'(x^2) = -\frac{\text{Ai}(0)}{2^{4/3}}. \tag{68}$$

Eq. (67) can be obtained directly from the integral representation (64), while Eq. (68) can be derived with the help of Eq. 2.16.33.1 of Ref. [49].

### D.2   The key integral with the Airy functions

Here we prove the identity

$$I(p,\tau) = \int_{-\infty}^{\infty} dx\, \text{Ai}(x^2)\text{Ai}'(px)\exp\left(\frac{2}{3}x^3\tau\right) = -\frac{\exp\left(\frac{p^3}{6}\frac{\tau}{(1-\tau^2)}\right)}{2^{1/3}3^{1/2}(1-\tau^2)^{1/6}} \text{Ai}\left(\frac{p^2}{2^{4/3}(1-\tau^2)^{2/3}}\right). \tag{69}$$

This is the key integral, which determines the PDF of the pendulum's angle and momentum in the absence of gravitation ($\omega = 0$), see Sec. C. It is absent in the standard tables of integrals [49, 50] and handbooks of special functions [48, 51].

### D.2.1 Canonization of the ternary cubic

Using the integral representation (64) and switching to imaginary $\tau$, we write $I(i\tau, p)$ as a triple integral:

$$
\begin{aligned}
I(i\tau, p) &= \int \mathrm{Ai}(x^2)\,\mathrm{Ai}'(px)\exp(2ix^3\tau/3)\,dx \\
&= \frac{1}{(2\pi)^2}\int dt\,ds\,dx\, is \exp i\left(\frac{t^3}{3} + x^2 t + \frac{s^3}{3} + xps + \frac{2}{3}x^3\tau\right).
\end{aligned}
\tag{70}
$$

As an auxiliary step, we rescale the $x$ variable:

$$
I(i\tau, p) = \frac{1}{(2\tau)^{1/3}}\frac{1}{(2\pi)^2}\int dt\,ds\,dx\, is \exp i\left(\frac{t^3}{3} + \frac{x^2 t}{(2\tau)^{2/3}} + \frac{s^3}{3} + \frac{xps}{(2\tau)^{1/3}} + \frac{x^3}{3}\right).
\tag{71}
$$

To bring the ternary cubic to a canonic form we make a linear transformation of the variables $x$ and $t$:

$$
\begin{pmatrix} x \\ t \end{pmatrix} = M\begin{pmatrix} u \\ v \end{pmatrix}, \qquad M = \frac{1}{(e^\theta + e^{-\theta})^{2/3}}\begin{pmatrix} e^{\theta/3}(e^\theta - e^{-\theta})^{1/3} & -e^{-\theta/3}(e^\theta - e^{-\theta})^{1/3} \\ e^{-2\theta/3} & e^{2\theta/3} \end{pmatrix},
\tag{72}
$$

where the angle $\theta$ is defined such that $\tau = \sinh\theta$. The Jacobian of the transformation is given by:

$$
J = \det M = (\tanh\theta)^{1/3} = \frac{\tau^{1/3}}{(1 + \tau^2)^{1/6}}.
\tag{73}
$$

Hence we get

$$
I(i\tau, p) = \frac{1}{2^{1/3}(1 + \tau^2)^{1/6}}\frac{1}{(2\pi)^2}\int ds\,du\,dv\, is \exp i\left(\frac{s^3 + u^3 + v^3}{3} + P(e^{\theta/3}u - e^{-\theta/3}v)s\right)
\tag{74}
$$

where

$$
P = \frac{p}{(e^\theta + e^{-\theta})^{2/3}} = \frac{p}{2^{2/3}(1 + \tau^2)^{1/3}}.
\tag{75}
$$

Comparing with Eq. (69), we see that it is equivalent to the following identity:

$$
\begin{aligned}
L(\tau, P) &\equiv \frac{1}{(2\pi)^2}\int ds\,du\,dv\, is \exp i\left(\frac{s^3 + u^3 + v^3}{3} + P(e^{\theta/3}u - e^{-\theta/3}v)s\right) \\
&= -\frac{1}{3^{1/2}}\exp\left(\frac{2P^3}{3}i\tau\right)\mathrm{Ai}(P^2),
\end{aligned}
\tag{76}
$$

that will be proven below.

### D.2.2 Auxiliary integral

Consider an integral

$$
K(x, y) \equiv \frac{1}{(2\pi)^2}\int du\,ds\,dv\, is \exp i\left(\frac{s^3 + u^3 + v^3}{3} + (xu + yv)s\right).
\tag{77}
$$

Its Fourier transform with respect to $x$ and $y$ can be evaluated easily:

$$
K(p, q) = \int dx\,dy\, e^{-ipx - iqy}K(x, y) = -\frac{2\pi}{3}J_0\left[\frac{2}{3}(p^3 + q^3)^{1/2}\right]\theta(p^3 + q^3).
\tag{78}
$$

Now we take the inverse Fourier transform:

$$K(x, y) = -\frac{2\pi}{3} \int \frac{dp\, dq}{(2\pi)^2}\, \theta(p+q) J_0\left[\frac{2}{3}(p^3+q^3)^{1/2}\right] e^{ipx+iqy}. \tag{79}$$

With $p+q=u$, $p-q=v$ and $s=(x+y)/2$, $r=(x-y)/2$ we get

$$K(x, y) = -\frac{1}{12\pi} \int_0^\infty du \int_{-\infty}^\infty dv\, J_0\left[\frac{1}{3}(u^3+3xv^2)^{1/2}\right] e^{isu+irv}. \tag{80}$$

We take the $v$-integral first:

$$V \equiv \int_{-\infty}^\infty dv\, J_0\left[\frac{1}{3}(u^3+3uv^2)^{1/2}\right] e^{irv} = \int_{-\infty}^\infty dv\, J_0\left[\frac{u^{1/2}}{3^{1/2}}(u^2/3+v^2)^{1/2}\right] \cos rv. \tag{81}$$

Writing $u^2/3+v^2=z^2$ and $a=u/\sqrt{3}$, we get

$$V = 2 \int_a^\infty dz \frac{z}{\sqrt{z^2-a^2}} J_0\left[\frac{u^{1/2}}{3^{1/2}} z\right] \cos r\sqrt{z^2-a^2}. \tag{82}$$

This is the integral 2.12.22.6 from Ref. [49]:

$$V = 2\frac{\cos[(u/\sqrt{3})\sqrt{u/3-r^2}]}{\sqrt{u/3-r^2}} \theta(u/3-r^2). \tag{83}$$

Hence Eq. (80) reduces to ($u=3r^2z$)

$$K(x, y) = -\frac{r}{2\pi} \int_1^\infty dz\, e^{3ir^2sz} \frac{\cos[\sqrt{3}r^3z\sqrt{z-1}]}{\sqrt{z-1}}. \tag{84}$$

Changing the variables $z=1+k^2$, we get

$$K(x, y) = -\frac{r}{\pi} \int_0^\infty dk\, e^{3ir^2s(1+k^2)} \cos[\sqrt{3}r^3k(1+k^2)] = -\frac{r}{2\pi} \int_{-\infty}^\infty dk\, e^{3ir^2s(1+k^2)+i\sqrt{3}r^3k(1+k^2)}. \tag{85}$$

Finally, eliminating the quadratic term ($k=(t-s)/r\sqrt{3}$), we arrive at the Airy-type integral

$$\begin{aligned} K(x, y) &= -\frac{1}{2\pi\sqrt{3}} \int_{-\infty}^\infty dt\, \exp i\left(\frac{t^3}{3}+t(r^2-s^2)+\frac{2}{3}\left(3r^2s+s^3\right)\right) \\ &= -\frac{1}{\sqrt{3}} \exp i\left(\frac{2}{3}\left(3r^2s+s^3\right)\right) \text{Ai}(r^2-s^2). \end{aligned} \tag{86}$$

In terms of original variables $x$ and $y$,

$$K(x, y) = -\frac{1}{\sqrt{3}} \exp\left(i\frac{x^3+y^3}{3}\right) \text{Ai}(-xy). \tag{87}$$

### D.2.3  Final step

Using Eqs. (76), (77) and (87), we obtain

$$L(\tau, P) = K(Pe^{\theta/3}, -Pe^{-\theta/3}) = -\frac{1}{\sqrt{3}} \exp\left(i\frac{2P^3\tau}{3}\right) \text{Ai}(P^2) \tag{88}$$

As discussed above, that completes the proof of Eq. (69).

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
