# Peer review of "Inverted pendulum driven by a horizontal random force: statistics of the never-falling trajectory and supersymmetry"

_SciPost Physics_

## Round 1 · Referee Report · Anonymous (Referee 1) · 2022-5-22

Strengths

The authors study statistical properties of a never falling trajectory in an inverted pendulum subject to a stochastic force. I was not familiar with this particular topic (of never falling trajectories) and therefore cannot comment on the novelty of the results. Given that the authors compared their analytic treatment with Monte Carlo simulations I trust that the results obtained using the developed Fokker-Planck approach are correct.

Weaknesses

Given my limited understanding of the particular mathematical problem of finding never falling trajectory I found discussion of its relevance to anything rather scarce. Because for each driving protocol such a trajectory is different it seems impossible to find it without a-priory knowledge of the driving force in full detail, at least the authors do not comment about that. In this sense it was not clear to me how existence of such a trajectory can help one to stabilize the system as the authors claim in introduction.

I believe the authors obtained an interesting mathematical result but I do not see how it can be relevant to any physical setup unless there is a discussion of how one can find such trajectory and how robust the initial conditions are.

Report

The paper probably merits acceptance as a mathematical result, however since math physics is not my area of expertise I cannot comment how novel and important the result is. It is not clear to me from the text how these results can be relevant to any physical setup.
  • validity: high
  • significance: good
  • originality: good
  • clarity: good
  • formatting: excellent
  • grammar: excellent

Author:  Mikhail Skvortsov  on 2022-06-30  [id 2622]

(in reply to Report 1 on 2022-05-22)

We thank the Referee for reading and accessing the manuscript, and for his/her criticism.

The main concern of the Referee, as mentioned in the Weaknesses section, is that the proposed approach is helpless in finding a particular NFT. This is indeed true: For a generic nonlinear differential equation, obtaining an explicit solution as a function of an arbitrary drive f(t) is impossible. However, in the case when the drive f(t) is random, one is typically interested in ensemble-averaged quantities. Such a statistic description of stochastic differential equations is in a sense analogous to thermodynamics, when instead of focusing on individual particle’s trajectories one deals with average characteristics.

For the pendulum problem considered in the present paper, we follow the same line and consider equal-time distribution function of the NFT’s angle and velocity.

Another concern of the Referee is the lack of relevance “to any physical setup”. In the previous version, we were mainly focusing on the pendulum problem, briefly mentioning in Conclusion its connection to other physical applications. Following Referee’s criticism, we extended the last part of Conclusion by discussing the relation to inhomogeneous superconductors (the last paragraph of Sec. 5). Here we explain that solving the Usadel equation with a random order parameter is similar to finding an NFT in the stochastic Whitney’s problem, albeit the spectral angle in the former problem is a complex rather than real variable. This example demonstrates that the stochastic Whitney’s problem is a prototypical problem for a number of related physical setups.

---

## Round 1 · Referee Report · Anonymous (Referee 2) · 2022-6-7

Strengths

The strength of the paper is that it clearly frames the problem and the generalization that it considers. The assumptions and framework are clearly outlined. The details are relegated to the appendices making the main text very readable.

Weaknesses

The weakness of the paper is that some of the assumptions made are not fleshed out clearly. For example, the uniqueness argument requires more details since the applicability of the supersymmetric formulation depends on it.

Secondly, the explicit connection between the problem considered here and that of the NFT is not obvious. This is because the formulation relies on the existence of NFT and the existence of NFT for the stochastic case is studied through this formulation.

Report

This paper considers the inverted pendulum problem subject to a gaussian uncorrelated random force. This is a stochastic generalization of Whitney's problem, which considered if there exists a non-falling trajectory for the inverted pendulum problem subject to a deterministic force f(t).

The paper is clearly written and the problem is framed well, however, the assumptions on the use of Parisi Sourlas framework and how the NFT is altered by the non-deterministic nature of the force are not very clear. The following are some of the questions, I would like the authors to consider reserve my recommendation based on the answers to the questions below.

1) Even for the deterministic case, the question of continuity (rod snapping back after being arbitrarily close to the floor) has plagued mathematicians. It is not very clear if this issue could be even more severe for the random driving force.

2) Following point one, the applicability of the Parisi-Sourlas formulation seems to be reliant on the existence of the NFT, although, it is not very clear if the existence of NFT is obvious for the stochastic forcing. This point needs to be addressed and if there are any caveats, they should be acknowledged.

3) The uniqueness of the NFT trajectory argument should be elaborated further if possible, since, it is used in dropping the absolute value operation in the functional integral.

4) The key aspect where the framework becomes the study of NFT is to convert the evolution problem into a boundary value problem. This a done by imposing the boundary conditions 11. Can the authors explain, in case there are no NFTs in the dynamics, how will it show up in their analysis?

5) As shown in figure 1, the NFT splits into diverging trajectories near the boundaries, how does this affect the validity of the Parisi-Sourlas formulation? It seems that this divergence of trajectories should prohibit the dropping of the absolute value in the determinant. The near boundary divergence effect seems to be spoiling the whole analysis in my opinion.

  • validity: good
  • significance: good
  • originality: good
  • clarity: high
  • formatting: excellent
  • grammar: excellent

Author:  Mikhail Skvortsov  on 2022-06-30  [id 2623]

(in reply to Report 2 on 2022-06-07)

We thank the Referee for his/her comments and questions. The text is improved following Referee’s criticism. One of the most valuable suggestions was to explain the arguments behind the uniqueness of a non-FT. We found this call very insighting and presented a proof in Appendix A of the revised manuscript.

1&2) The fact that we consider random drive actually does not spoil the mathematical proof of the existence of the non-FT. Ensemble averaging over realization of the driving force we perform actually means that first the non-FT should be determined for a particular f(t) dependence. According to mathematicians, such a trajectory does exist. Then disorder averaging should be done by sampling over realizations of f(t). Statistical property of the non-FT, the object well-defined for any f(t), is the subject of our study. Hence, mathematical issues do not become more severe in the nondeterministic case.

3) Indeed, the uniqueness of the non-FT is a crucial observation for the whole business. We fully agree with the Referee that this important issue was not properly discussed. We respond by adding a one-page Appendix A, where the proof of the uniqueness is provided.

4) The NFT manifests itself in the very existence of the zero mode of the transfer-matrix Hamiltonian. For other stochastic differential equations where an NFT may not exist the Hamiltonian does not have a zero mode. We add a special paragraph discussing this relation (the second paragraph of Sec. 2.3).

5) Figure 1 shows many non-falling solutions, but they are solutions of different boundary-value problems specified by the angles at t=0 and t=T. As long as these values are given, the solution is unique, as discussed in the text. Therefore, there is no problem with multiple non-FT and hence with the sign of the determinant.

The apparent divergence of different non-FT for different boundary conditions seen in Fig. 1 is in fact related to the Lyapunov exponent, which measures the loss of different-time correlations at the non-FT. We add a paragraph in Conclusion to address this issue (the second paragraph of Sec. 5).

---

## Editorial Decision

resubmitted